# Sigma-2 Receptor Ligand Binding Modulates Association between TSPO and TMEM97

**DOI:** 10.3390/ijms24076381

**Published:** 2023-03-28

**Authors:** Bashar M. Thejer, Vittoria Infantino, Anna Santarsiero, Ilaria Pappalardo, Francesca S. Abatematteo, Sarah Teakel, Ashleigh Van Oosterum, Robert H. Mach, Nunzio Denora, Byung Chul Lee, Nicoletta Resta, Rosanna Bagnulo, Mauro Niso, Marialessandra Contino, Bianca Montsch, Petra Heffeter, Carmen Abate, Michael A. Cahill

**Affiliations:** 1School of Dentistry and Medical Sciences, Charles Sturt University, Wagga Wagga, NSW 2678, Australia; 2Research and Development Department, The Ministry of Higher Education and Scientific Research, Baghdad 10065, Iraq; 3Department of Science, University of Basilicata, Viale dell’Ateneo lucano 10, 85100 Potenza, Italy; 4Department of Pharmacy-Drug Sciences, University of Bari ‘ALDO MORO’, Via Orabona 4, 70125 Bari, Italy; 5Life Sciences and Health, Faculty of Science, Charles Sturt University, Wagga Wagga, NSW 2650, Australia; 6School of Medicine and Psychology, Australian National University, Florey Building, 54 Mills Road, Acton, ACT 2601, Australia; 7Department of Radiology, Perelman School of Medicine, University of Pennsylvania, Philadelphia, PA 19104, USA; 8Department of Nuclear Medicine, Seoul National University Bundang Hospital, Seoul National University College of Medicine, Seongnam 13620, Republic of Korea; 9Center for Nanomolecular Imaging and Innovative Drug Development, Advanced Institutes of Convergence Technology, Suwon 16229, Republic of Korea; 10Dipartimento di Medicina di Precisione e Rigenerativa e Area Jonica (DIMePRe-J), Università degli Studi di Bari ‘ALDO MORO’, Piazza Giulio Cesare, 70124 Bari, Italy; 11Center for Cancer Research and Comprehensive Cancer Center, Medical University of Vienna, Borschkegasse 8a, 1090 Vienna, Austria; 12Consiglio Nazionale delle Ricerche (CNR), Istituto di Cristallografia, Via Amendola, 70125 Bari, Italy; 13ACRF Department of Cancer Biology and Therapeutics, The John Curtin School of Medical Research, The Australian National University, Canberra, ACT 2601, Australia

**Keywords:** Sigma-2 receptor, TSPO, TMEM97, PGRMC1, protein-protein interaction

## Abstract

Sigma-2 receptor (S2R) is a S2R ligand-binding site historically associated with reportedly 21.5 kDa proteins that have been linked to several diseases, such as cancer, Alzheimer’s disease, and schizophrenia. The S2R is highly expressed in various tumors, where it correlates with the proliferative status of the malignant cells. Recently, S2R was reported to be the transmembrane protein TMEM97. Prior to that, we had been investigating the translocator protein (TSPO) as a potential 21.5 kDa S2R candidate protein with reported heme and sterol associations. Here, we investigate the contributions of TMEM97 and TSPO to S2R activity in MCF7 breast adenocarcinoma and MIA PaCa-2 (MP) pancreatic carcinoma cells. Additionally, the role of the reported S2R-interacting partner PGRMC1 was also elucidated. Proximity ligation assays and co-immunoprecipitation show a functional association between S2R and TSPO. Moreover, a close physical colocalization of TMEM97 and TSPO was found in MP cells. In MCF7 cells, co-immunoprecipitation only occurred with TMEM97 but not with PGRMC1, which was further confirmed by confocal microscopy experiments. Treatment with the TMEM97 ligand 20-(*S*)-hydroxycholesterol reduced co-immunoprecipitation of both TMEM97 and PGRMC1 in immune pellets of immunoprecipitated TSPO in MP cells. To the best of our knowledge, this is the first suggestion of a (functional) interaction between TSPO and TMEM97 that can be affected by S2R ligands.

## 1. Introduction

The sigma-2 receptor (S2R) is an 18–21 kDa membrane protein that is a potential biomarker of the proliferative status of solid tumors [1,2,3]. Many S2R ligands modulate disease states [3,4,5] and can induce cell death [6], leading to an interest in their therapeutic potential [1,3,7]. Although the sigma-1 receptor (S1R) was successfully cloned in 1996 [8] and crystallized in 2016 [9], cloning and structural characterization of S2R has proved more elusive. Major progress was made when a photoactivated S2R ligand became cross-linked to PGRMC1, identifying a PGRMC1-containing complex with S2R activity [10]. This led to considerable controversy [11,12,13,14,15,16] over the non-identity of PGRMC1 with S2R after some papers referred to PGRMC1 as “PGRMC1/S2R”, assuming that PGRMC1 was S2R (e.g., [17,18,19]).

In 2017, the transmembrane protein 97 (TMEM97), which is also known as meningioma-associated protein 30 (MAC30), was identified as a gene coding for S2R activity in tumor cell lines [20]. Attenuation of TMEM97 by siRNA in PC-12 cells resulted in a proportional reduction of S2R ligand binding. Although overexpression of TMEM97 in Sf9 insect cells caused an increase of saturable [^3^H]DTG binding, the same was not true of PGRMC1 [20]. Mach and colleagues apparently resolved the controversy of how TMEM97 could be the S2R [20] by showing that TMEM97 forms a complex with PGRMC1 that directs enhanced internalization of the low-density lipoprotein receptor (LDLR), thereby revealing that TMEM97 could participate in protein-protein interactions [21]. In 2021, separate ligand-bound crystal structures were solved for TMEM97 with two respective known ligands, which enabled the structural prediction of hundreds of novel S2R ligands: two of which were also solved as ligand-bound crystal structures [22].

Prior to the 2017 publication that S2R activity was associated with TMEM97 [20], we had been independently investigating the translocator protein (TSPO) as a potential 18–21 kDa candidate S2R protein. In a preliminary PGRMC1-HA co-immunoprecipitation (co-IP) and mass spectrometric protein identification assay, TSPO was identified as a potential PGRMC1-interacting protein among a list of many identified proteins, so we hypothesized that could be associated with S2R activity in MIA PaCa-2 (MP) pancreatic adenocarcinoma cells. Consequently, we hypothesized that TSPO expression could be associated with S2R activity in MP cells. TSPO, also known as the peripheral benzodiazepine receptor (PBR), is a mitochondrial transmembrane protein with reported heme- and cholesterol-binding activities [23], which is thought to be a critical cholesterol transporter [24]. The previous association of TSPO with steroidogenesis has been stringently criticized [25]. Once TMEM97 was identified as S2R, we hypothesized that TSPO could be its interactor, as shown for PGRMC1 [21]. Using the nanomolar-affinity (S2R *K*_i_ = 11 nM) and green-emitting fluorescent S2R ligand SW120 [10], we started our investigation on the possible interrelation between TMEM97 and TSPO in MP cells. The data obtained prompted us to evaluate the presence of TMEM97 and TSPO, in a panel of functionally different cell lines, by saturation analysis with the appropriate radioligands. In agreement with previous data reporting higher density of S2R in several lines of breast and pancreatic cancer cells than in the normal (immortalized) cells [26,27,28], pancreatic adenocarcinoma MP and breast adenocarcinoma MCF7 cells were selected because of the density of the TMEM97 and TSPO proteins. Thus, in MP and MCF7, we assessed the association possibility between TMEM97, PGRMC1, and TSPO by proximity ligation assay (PLA), co-IP, and fluorescent staining. Note that all the assays employed (PLA, co-immunofluorescence, or co-IP) can detect the presence of proteins that are bound in higher-order protein complexes. Therefore, none of them can prove direct physical contact between the two respective proteins, although PLA epitopes from two target proteins must be within 40 nm to obtain a signal [29].

In MCF7 cells, the association between TMEM97 and TSPO but not between TSPO and PGRMC1 was detected. In MP cells, which exhibited the highest abundance of TSPO, co-IP also demonstrated the association between PGRMC1 and TSPO. Importantly, 20-(*S*)-hydroxycholesterol (20S-OHC) has recently been proposed as an endogenous TMEM97 ligand [30]. Thus, we studied its effects, together with the impact of other small TMEM97-interacting molecules, on the TSPO-TMEM97 association. These data together suggest context/cell-dependent interactomes for TSPO and TMEM97 that can modulate S2R ligand-binding/activity.

## 2. Results

### 2.1. TSPO and TMEM97 Are Bound by the S2R Ligand SW120 in MP Cells

We preliminary investigated whether TSPO could influence the binding of S2R ligands to cells using the fluorescent S2R ligand SW120. We attenuated TMEM97 by siRNA or TSPO levels by stably transfecting MP cells with shRNA-expressing lentivirus and compared the reduction in SW120 binding in the TMEM97- or TSPO-attenuated cells (Figure 1). TMEM97 mRNA was reduced by approximately 80%, as assayed by RT-PCR (Figure 1A). This was accompanied by an approximately 20% reduction in the binding of SW120 to MP cells, as assayed by the detection of the median fluorescence intensity (MFI) of cells separated by flow cytometry (Figure 1B). Knockdown of TSPO mRNA in MP cells containing stably transduced lentiviral shRNA resulted in approximately 70% reduction in TSPO shRNA cells relative to scrambled shRNA (Scr shRNA) control cells (Figure 1C). The protein level knockdown of TSPO is shown in Appendix A–E. This led to an approximately 50% reduction of SW120-dependent MFI (Figure 1D). SW120 binding to S1R in the MP cells did not significantly contribute to the SW120 fluorescence observed because cells were also incubated with 1.8 µM SKF10.047 to block S1R, compared to solvent DMSO as a control. This level of SKF10.047 is reported to block S1R so that S1R did not contribute to the SW120-generated signal in accordance with the S2R versus S1R selectivity of SW120 [31,32].

### 2.2. TSPO and TMEM97 Affect the Same SW120-Binding Sites in MP Cells

In the above results, SW120 binding was influenced by levels of both TSPO and TMEM97. This could be due to the same molecule binding two different binding sites at each protein, in which case, we would expect to see additive effects upon simultaneous attenuation. Alternatively, both proteins could be affecting the same binding sites, in which case, the effects of a double knockout (DKO) should be about the same as each individual knockout. To better understand the effect posed by both proteins, we generated DKO MP cells using shRNA attenuation for TSPO and transiently transfected siRNA for TMEM97. The attenuation of each protein caused approximately equal reductions in SW120 binding. Strikingly, the DKO cells showed no additive reduction in signal strength (Figure 2, left-hand bars). The lack of an additive effect on SW120 binding upon simultaneous attenuation of both TSPO and TMEM97 suggests that TSPO and TMEM97 levels affect the same S2R ligand-binding sites.

DTG exhibits binding to both S1R and S2R, whereas the S2R antagonist RHM-1 has 300-fold specificity for S2R relative to S1R and can be used in binding competition assays to discriminate between S1R and S2R binding [33,34]. Upon addition of either DTG or RHM-1, all the cell lines exhibited a significant reduction in SW120 binding (Figure 2), indicating that the SW120 binding to MP cells, as measured by flow cytometry MFI, was at least mostly due to the S2R. Following the rationale of Weng et al. [34], who argue that RHM-1 is a superior selective ligand for S2R over DTG, these results indicate that the SW120 fluorescence observed by flow cytometry MFI values in Figure 2 can be attributed to specific SW120-S2R complexes. While the residual signal after RHM-1 treatment may be due to the unspecific binding of SW120, taken together, these results suggest that TSPO and TMEM97 contribute cooperatively to SW120 binding to the S2R. This suggests that either the binding site is formed upon the interaction between these two proteins or that the close proximity between the two proteins may allosterically stabilize the S2R binding site. Further research will be required to characterize this situation.

In control experiments, we confirmed that all respective antibodies gave positive immune fluorescence signals, which were attenuated by the knockdown of PGRMC1 or TSPO, respectively (Appendix A).

### 2.3. TSPO Colocalizes with TMEM97 in MP Cells

The results from the SW120 experiments prompted us to perform proximity ligation assays in MP cells to evaluate the close proximity of the proteins. Thus, we evaluated the following interactions: PGRMC1 + TMEM97, PGRMC1 + TSPO, and TSPO + TMEM97. We obtained positive PLA signals for the TMEM97/TSPO antibody pair (Figure 3). These results are consistent with TSPO and TMEM97 forming a protein complex, which is possibly required for S2R activity in these cells. We observed low levels of PLA signal between PGRMC1 and TMEM97. Differences to the higher levels observed by Riad et al. in HeLa cells [21] may reflect cell-specific differences. Riad et al. also used a polyclonal anti-PGRMC1 antibody, whereas we employed an anti-HA tag antibody against the HA tag of exogenously expressed PGRMC1-HA. It is additionally conceivable that the TMEM97/PGRMC1 interaction is sensitive to the presence of the HA tag at the PGRMC1 C-terminus. No positive PLA signals for the PGRMC1/TSPO antibody pair were found. However, taken together, our results show a clear and previously undescribed interaction between TSPO and TMEM97. Since the TMEM97-PGRMC1 interrelation has been previously established and could be cell type-specific, but the TSPO-TMEM97 interaction was novel, we concentrated further efforts on the latter.

### 2.4. Saturation Analyses by Radioligand Binding to Detect TMEM97 and TSPO in Functionally Different Cell Lines

Four cell lines were then selected to identify the most appropriate cell models for the protein-protein TSPO-TMEM97 interaction study, which may be context-dependent. Rat glioma C6 cells are routinely used in TSPO-binding experiments [35,36,37] and express S2R [38]. By contrast, colorectal adenocarcinoma LoVo cells were previously reported to present S2R with a low density [28]. MP and MCF7 cells both display S2R, with the latter cell being employed as a model to study S2R-mediated activity [39,40,41,42]. We also generated an MCF7 subline (shMCF7TMEM97), which exhibited lower TMEM97 protein levels due to shRNA knockdown (see paragraph below). As depicted in Table 1 and Appendix A, saturation analysis detected the presence of receptors for TSPO ligands in all studied cell lines to different extents. The highest amount was detected in MP cells (B_max_ = 7.3 pmol/mg of protein) and the lowest in MCF7 cells (B_max_ = 1.02 pmol/mg of protein).

Analogously, S2R ligand binding was observed in all the selected cells except for LoVo. The highest level was measured in MCF7 and C6 cells (B_max_ = 2.02 and 2.15 pmol/mg of protein, respectively) (Table 1, Appendix A). It is noteworthy that receptors for ligands binding TSPO and S2R are expressed in the same cell lines (except for LoVo cells) and that the reduction in S2R ligand-binding observed in shMCF7TMEM97 cells (B_max_ = 0.891 pmol/mg of protein) is accompanied by an increase in TSPO ligand-binding (B_max_ = 2.06 pmol/mg of protein) compared with the MCF7 parental line, suggesting a compensatory effect that is consistent with a hypothesized functional link existing between the expression patterns of TMEM97 and TSPO.

### 2.5. Generation of MCF7 and MP Cells with Attenuated Levels of TMEM97/S2R

With the aim of generating models to study the protein-protein interaction in more detail in the two selected cell lines, MCF7 and MP cells, we attenuated the level of TMEM97. In MP cells, TMEM97 levels were reduced by siRNA, as described by Alon et al. [20], and the level of gene expression was measured by RT-PCR. In both 10 nM and 50 nM siRNA concentrations, TMEM97 mRNA was significantly reduced by around 80% (*t*-test *p* < 0.0001) (Figure 1A), leading to significantly reduced SW120 binding (*t*-test *p* < 0.0001) (Figure 1B). Here, we also included an S1R-specific competitive control, SKF10.047, to ensure the S2R specificity of SW120 binding.

In MCF7 cells, TMEM97 levels were reduced by shRNA, and the amount of residual TMEM97 was measured by saturating [^3^H]DTG (Table 1, B_max_ = 0.891 pmol/mg protein in shMCF7TMEM97 cells vs. B_max_ = 1.795 pmol/mg protein in WT cells; Appendix A, respectively). As recently reported, a corresponding significant reduction in the binding of two structurally different green-emitting S2R-specific fluorescent ligands (i.e., F412 and NO1) was detected [43].

### 2.6. TMEM97 Co-Immunoprecipitates with TSPO

In order to support the potential protein-protein associations between TSPO and TMEM97 and/or PGRMC1, we performed co-IP experiments with a specific anti-TSPO primary antibody. In MCF7 cells, we found that TMEM97 was detected in immune pellets obtained by precipitation of TSPO but that PGRMC1 was not (Figure 4A,B). The co-immunoprecipitation data in MCF7 cells were consistent with confocal microscopy results in the same cells, where colocalization was observed between a fluorescent S2R ligand and TSPO (Appendix A) but not with PGRMC1, which exhibited a nuclear localization (Appendix A). TSPO and PGRMC1 were immunostained by specific antibodies against each protein, while S2R (as a surrogate for TMEM97) was stained via a recently reported CY5-labeled S2R-specific ligand [43].

As expected, the intensity of a TMEM97-related band was greater in parental MCF7 than in shTMEM97MCF7 cells (Figure 4A). In both MCF7-based cell lines, no signal was found when immune complexes precipitated with anti-TSPO were blotted with anti-PGRMC1 antibody (Figure 4B). By contrast, in MP cells, we observed that both TMEM97 (Figure 4C) and PGRMC1 (Figure 4D) co-immunoprecipitated with TSPO. This suggests the existence of a possible multiprotein complex formed by TMEM97, TSPO, and PGRMC1 in MP cells. Alternatively, TSPO could be present in separate protein complexes containing either PGRMC1 or TMEM97, despite the absence of a positive PGRMC1/TSPO PLA signal.

To illuminate the difference observed between the two tumor cell lines, we performed Western blot experiments of whole cell lysates to evaluate the expression levels of TMEM97 and PGRMC1. TMEM97 was detected at higher levels in MCF7 than in MP cells (Figure 4E), confirming the saturation analyses performed with the S2R radioligand DTG (Table 1). PGRMC1 was detected at higher levels in MP than in MCF7 cells (Figure 4F). Therefore, more abundant expression of PGRMC1 in MP than in MCF7 cells could explain the detected association between TSPO and PGRMC1 in MP but not in MCF7 cells.

### 2.7. 20S-OHC Reduced the Associations TSPO-TMEM97 and TSPO-PGRMC1 in MP Cells

MP cells were treated for 24 h with 20S-OHC, which was recently proposed as an endogenous TMEM97 ligand [30], and immunoprecipitated with an anti-TSPO antibody. The analysis performed via Western blot revealed that 20S-OHC reduced the presence of both TMEM97 and PGRMC1 in TSPO immune precipitates (Figure 5). In fact, both TMEM97 (Figure 5A, upper panel) and PGRMC1 (Figure 5B, lower panel) protein bands were lower by more than half in 20S-OHC treated cells than in untreated control cells.

### 2.8. Effect of Siramesine Oxalate Salt and FA10 Hydrochloric Salt on TSPO Interactions-With TMEM97 and PGRMC1 in MP Cells

We thus investigated the role of synthetic TMEM97/S2R ligands on these interactions. MP cells were treated for 2 h with 10 μM siramesine oxalate salt (S) [44] or FA10 ([1-(3-(6,7-dimethoxy-3,4-dihydroisoquinolin-2(1*H*)-yl)propyl]-5-methoxy-1,2,3,4-tetrahydroquinoline hydrochloric salt) [45], immunoprecipitated with an anti-TSPO antibody and analyzed via Western blotting with specific antibodies directed against TMEM97 (Figure 6A) and PGRMC1 (Figure 6B). Notably, the former compound is a reference S2R agonist, whereas the latter is a novel, highly selective S2R ligand devoid of antiproliferative activity (in contrast to siramesine). Siramesine oxalate salt significantly increased the interactions between TSPO-TMEM97 (Figure 6A) and TSPO-PGRMC1 (Figure 6B) in comparison with untreated MP cells. In the presence of FA10 hydrochloric salt, the interaction between TSPO and PGRMC1 was stronger than in control cells (Figure 6B), but no significant changes were observed with respect to the TSPO-TMEM97 interaction (Figure 6A).

## 3. Discussion

For many years S2R was known as an enigmatic S2R ligand-binding site, which was attributed to protein species of 21.5 kDa [3]. Before the report that TMEM97 binds S2R ligands [20], we had reasoned that TSPO (predicted mass 18.8 kDa) could be post-translationally modified to possibly generate the 21.5 kDa species, which is why we were investigating TSPO. After Alon et al. [20] reported that TMEM97 binds to S2R ligands, we attempted to reconcile our results with theirs. As shown above, we observed proximal and functional interactions between TMEM97 and TSPO, as well as with the previously reported TMEM97 interaction partner PGRMC1 [21].

TMEMs are upregulated in cancers, where they are associated with tumor progression, invasion, and metastasis [46]. Also known as MAC30, TMEM97, is an endoplasmic reticulum-resident membrane protein [20]. TMEM97 was identified as a cholesterol-regulating gene by functional RNAi screening [47]. We show that both TSPO and TMEM97 are required for SW120 binding. Importantly, attenuation of each protein in the double knockdown did not cause an additive reduction of the SW120 signal, which would be expected if SW120 independently binds the two proteins. These results, suggesting that these proteins form a complex, prompted further investigations that we conducted in two cell line models selected for their TMEM97/S2R and TSPO expression patterns, i.e., MCF7 and MP cells. Due to the previously detected interaction between TMEM97 and PGRMC1 in protein complexes, also PGRMC1 was included in the investigation. Results from PLA and co-IP experiments support an interaction between TMEM97 and TSPO in both cell lines. In co-IP pellets of TSPO immunoprecipitates, no PGRMC1 was detected in MCF7 cells, in agreement with the lack of colocalization of the two proteins, as detected by confocal microscopy. On the other hand, PGRMC1 was detected in co-IP pellets of TSPO immunoprecipitates in MP cells, in contrast to the PLA results in MP cells, where PGRMC1-TSPO interaction was not detected. Taken together, these results suggest that the interaction of TSPO and PGRMC1 does not correspond to a close proximity revealable by PLA. Alternatively, an inappropriate orientation could hamper the annealing of the primers to generate a PLA signal, or TSPO interaction may block the antigen for the PGRMC1 antibody or *vice versa*. Notably, the differences observed between the two cells for co-IP partners suggest a context-dependent interactome that may also reflect the density of the proteins.

Overall, all the performed experiments together suggest protein-protein interactions that involve different partners in different cells: TSPO + TMEM97 in MCF7 and TSPO + TMEM97 and TSPO + PGRMC1 in MP cells. These interactions appear to be functional. The administration of endogenous (20S-OHC) or exogenous (siramesine and FA10) TMEM97/S2R ligands affects the co-IP profile, suggesting still unexplored pathways for S2R ligands-mediated activity.

The nature of the TMEM97-TSPO association remains unclear. We do not know whether the two proteins are in a direct physical complex or are merely closely juxtaposed in a common protein complex. The presence of a positive PLA signal argues for at least intimate proximity.

In HeLa cells, it has been demonstrated that PGRMC1 and TMEM97 are present in the complex bound by S2R ligands because a photoactivatable S2R ligand bound to the S2R formed a covalent cross-link with PGRMC1 upon photo-activation [10], and the two proteins colocalize in a protein complex as demonstrated [21,48]. Furthermore, we know that the TMEM97-binding site for S2R ligands is situated within the lipid bilayer in cells [22]. Finally, recognized TSPO ligands are pharmacologically distinct from S2R ligands [49].

Given these premises, we reason that TSPO may allosterically facilitate S2R ligand binding. This could be under a model where the TSPO-TMEM97 complex facilitates a TMEM97 conformation amenable to PGRMC1 binding (in MP cells) and/or to S2R ligand binding, thereby facilitating a change in TMEM97-complex status. In either model, TSPO could either remain present as part of the TMEM97-containing complex or be dissociated by allosteric protein conformational changes upon altered TMEM97-complex status. In the latter scenario, the dissociated TMEM97-containing component and/or the TSPO-containing component could be free to perform subsequent downstream functions (signal propagation) that were unavailable to the intact TSPO-TMEM97 complex (analogously to the ability of G_α_ and G_βγ_ subunits to perform downstream functions after their dissociation following ligand engagement by G-protein-coupled receptors [50]). However, note that Figure 5 suggests that TSPO may dissociate from both TMEM97 and PGRMC1 upon binding of the endogenous 20S-OHC S2R ligand. All the above scenarios remain hypothetical and will require subsequent experimental investigation.

TSPO is primarily reported as a mitochondrial protein [51], although it has been reported in the nucleus of spermatogenic cells [52] and in urinary exosomes [53], which implies that occupancy of the post-endosomal cytoplasmic vesicle compartment is at least sometimes possible. Future studies will need to address the subcellular location of the TMEM97/TSPO complex reported in this present study.

It is notable that all three proteins under consideration (TMEM97, TSPO, and PGRMC1) have interrelated biological attributes. TSPO is involved with steroid and heme biology [49] (which is the reason we began examining it), as is PGRMC1 [21,48,54,55], whereas TMEM97 is involved in regulating cholesterol levels [47,56], and its endogenous ligand appears to be 20S-OHC [30].

PGRMC1 is the archetypal member of the membrane-associated progesterone receptor (MAPR) family. This eukaryotic family forms a distinct sub-class of the cytochrome b5 domain superfamily, characterized by tyrosinate-mediated heme chelation, as opposed to the His-2 heme chelation of classical cytochrome b5 proteins [57,58]. Intriguingly, a group of bacterial proteins from the candidate phyla radiation (CPR) bacteria shares tyrosinate heme chelation with MAPR proteins. These CPR bacterial proteins have been named cytb_5MY_ proteins (MAPR-like cytochrome b5 with tyrosine/Y-heme chelation). It is unclear whether eukaryotic MAPR proteins originated from a CPR bacterial cytb_5MY_ protein or whether the cytb_5MY_ proteins arose by horizontal gene transfer of an MAPR gene into a CPR bacterium [58]. However, in the context of a putative biological relationship between mammalian PGRMC1 and TSPO, it is intriguing that the reconstructed ancestral CPR bacterial operon containing a cytb_5MY_ protein appears to have also contained a TSPO gene, as well as two other cytochrome b5 proteins and a putative ferric-reductase enzyme. The operon also contained a two-component inducible element, and all proteins possessed at least one trans-membrane helix, possibly suggesting that the operon encoded a redox-related inducible membrane complex [58]. All of those proteins have heme prosthetic groups [58,59] if we include the heme-associated TSPO [12]. While the CPR bacterial TSPO genes are less resemblant to eukaryotic TSPOs than other prokaryotic TSPOs, and therefore eukaryotic TSPO is not of CPR bacterial origin, it is possible that the colocation of TSPO and an MAPR-like cytb_5MY_ protein in CPR bacterial operons is reflective of a functional relationship between these two protein classes, possibly related to heme biology and redox functions. This remains highly conjectural yet provides hypotheses that may usefully direct future research.

## 4. Methods

### 4.1. Cell Culture

The identities of the two cell lines selected as a model to study the TMEM97-TSPO-PGRMC1 protein-protein interaction landscape, i.e., MCF7 and MP (MIA PaCa-2) were verified by Short Tandem Repeat (STR) profiling. Genomic DNA was extracted from these cell lines using the QIAamp Mini Kit (Qiagen, Hilden, Germany), according to the manufacturer’s instructions, and quantified on a BioSpectrometer Plus (Eppendorf, Hamburg, Germany). One nanogram of DNA extracted was amplified using the AmpFlSTR NGM Select kit (Applied Biosystems, Waltham, MA, USA) that simultaneously amplifies the same 16 loci (D3S1358, vWA, D16S539, D2S1338, D8S1179, D21S11, D18S51, D19S433, TH01, FGA, Amelogenin, D10S1248, D22S1045, D2S441, D1S1656, D12S391, SE33). Reactions were performed in C1000 thermal cycler (Bio-Rad, Hercules, CA, USA) following the manufacturer’s recommendations (Applied Biosystems NGM Manual). PCR products were separated using an ABI310 (Applied Biosystems) following the manufacturer’s recommendations. Samples have been prepared using 1 µL PCR or the NGM Allelic ladder mix (Applied Biosystems), 0.5 µL of GeneScanTM-500 LIZ internal Size Standard (Applied Biosystems), and 23.5 µL of deionized Hi-DiTM formamide (Applied Biosystems), denatured at 95 °C for 3 min and placed on ice for 3 min. Samples amplified using the NGM SElectTM kit were injected for 5 s at 15 kV and separated electrophoretically in Performance Optimized Polymer (POP-4TM; Applied Biosystems) using the GS STR POP4 (1 mL) G5 v2.MD5 Module (Applied Biosystems) and a 30 min run time. The genetic profiles were determined using Data Collection v 3.1 and Gene Mapper ID software v 3.2 (Applied Biosystems) with a 50 relative fluorescent unit (RFU) peak amplitude threshold for all dyes. The effectiveness of autosomal STR typing is reported in Appendix A.

Cells were maintained in complete media that contained a high glucose Dulbecco’s modified Eagle’s medium (DMEM) (Sigma-Aldrich, St. Louis, MO, USA, D5796), 10% fetal bovine calf serum (FBS, Sigma-Aldrich, F9423), and 1% penicillin-streptomycin (Sigma-Aldrich, P4333). Cells were incubated at 37 °C and 5% CO_2_ in a 150i CO_2_ incubator (Thermo Fisher Scientific, Heracell, Lane Cove, NSW, Australia) and passaged every time they reached 70% confluence (4 days). The rat C6 glioma cells were grown in HAM’S F12 with 10% heat-inactivated FBS, 100 U/mL penicillin, 100 μg/mL streptomycin, and 2 mM *L*-glutamine in a humidified atmosphere with 5% CO_2_ at 37 °C. MCF7 and shMCF7TMEM97 cells were grown in DMEM with 10% heat-inactivated FBS, 100 U/mL penicillin, 100 μg/mL streptomycin, and 2 mM *L*-glutamine in a humidified atmosphere with 5% CO_2_ at 37 °C. LoVo cells were grown in Ham’S F12 medium with 10% heat-inactivated FBS, 100 U/mL penicillin, 100 μg/mL streptomycin, and 2 mM *L*-glutamine in a humidified atmosphere with 5% CO_2_ at 37 °C.

### 4.2. Saturation Binding Assay

#### 4.2.1. Membrane Preparation

Membranes from C6 glioma were prepared as described by Moon et al. [35]. Briefly, rat C6 glioma was cultured to 80% confluence, the medium was removed, and the cells were scraped into PBS (pH = 7.2). After detaching, the cells suspended in PBS were homogenized with a Brinkman Polytron (setting 5 for 3 × 15 s). The homogenate was centrifuged at 37,000× *g* for 30 min at 4 °C, and the supernatant was discarded. The final pellet was resuspended in ice-cold 10 mM PBS (pH 7.2) and stored at −80 °C until use. Membranes from MCF7, shMCF7TMEM97, and LoVo cells were prepared as described by Abate et al. in 2012 (22890883). Briefly, the tumor cells were cultured to 80% confluence, the medium removed, and the cells rinsed in PBS. After detaching, the cells were suspended in ice-cold 10 mM TRIS-HCl (pH 7.4) buffer solution containing 0.32 M sucrose and homogenized in a Potter–Elvehjem homogenizer (Teflon pestle, Thermo Fisher Scientific). The homogenate was centrifuged at 31,000× *g* for 15 min at 4 °C, and the supernatant was discarded. The final pellet was resuspended in ice-cold 10 mM TRIS–HCl (pH 7.4) buffer solution and stored at −80 °C until use.

#### 4.2.2. Saturation Binding Assay by Radioligand at TSPO

In 0.5 mL of incubation buffer (PBS, pH 7.2), 100 µg of tumor cells membranes, different concentrations of the radioligand [^3^H]PK 11195, in the absence (total binding) and in the presence of 10 µM PK11195 as reference compound (non-specific binding) were suspended. The samples were incubated for 90 min at 25 °C. The incubation was stopped by rapid filtration on Whatman GF/C glass microfiber filters (pre-soaked in 0.3% polyethylenimine for 20 min). The filters were washed with 3 × 1 mL of ice-cold buffer (PBS, pH 7.2). The dissociation constant *K*_D_ and the receptor density Bmax were measured by the nonlinear fitting of the specific binding vs. the radioligand concentration using Prism software.

#### 4.2.3. Saturation Binding Assay by Radioligand at TMEM97/Sigma-2 Receptor

In 0.5 mL of incubation buffer (50 mM TRIS, pH 8), 400 µg of tumor cells membranes, different concentrations of the radioligand [^3^H]DTG, 1 µM (+)-pentazocine to mask sigma-1 subtype, in the absence (total binding) and in the presence of 10 µM DTG as reference compound (non-specific binding) were suspended. The samples were incubated for 120 min at 25 °C. The incubation was stopped by rapid filtration on Whatman GF/B glass microfiber filters (pre-soaked in 0.5% polyethylenimine for 60 min). The filters were washed with 3 × 1 mL of ice-cold buffer (50 mM TRIS, pH 7.4). The dissociation constant *K*_D_ and the receptor density Bmax were measured by the nonlinear fitting of the specific binding vs. the radioligand concentration using Prism software.

#### 4.2.4. Materials

[^3^H]PK1195 and [^3^H]DTG were purchased from Perkin-Elmer Life Sciences (Boston, MA, USA). (+)-Pentazocine and PK1195 were purchased from Sigma Aldrich (St. Louis, MO, USA); DTG was from Tocris. The K_D_ values and SEM were obtained using non-linear curve fitting (Prism v. 3.0, GraphPad, San Diego, CA, USA).

### 4.3. Co-Immunoprecipitation

Cells (1 × 10^7^), seeded in 10 cm dishes, were washed with PBS before lysis in 1 mL of cold low-stringency lysis buffer (100 mM NaCl, 10% glycerol, 50 mM HEPES pH 7.5, 1 mM EDTA, 0.5% NP-40, 10 μg/mL aprotinin, 1 mM phenylmethylsulfonyl fluoride, 1 μg/mL leupeptin, and 1 mM sodium orthovanadate) and treated as previously reported [60]. Briefly, the cell lysate was kept on ice for 15 min before centrifugation at 12,000 g for 10 min at 4 °C. The supernatant was collected, and anti-TSPO primary antibody (ab109497, Abcam, Cambridge, MA, USA) and 40 μL A/G PLUS agarose beads (SC-2003, Santa Cruz Biotechnology, Santa Cruz, CA, USA) were added. Following the incubation at 4 °C on a rocking platform for 2 h and centrifugation at 400 g for 2 min at 4 °C, the beads were washed three times with a 1 mL low-stringency lysis buffer pelleting between each wash at 2800 g for 3 min at 4 °C. The immune complexes were eluted with 50 μL of Laemmli sample buffer and analyzed by Western blotting.

### 4.4. Western Blotting

Cells were maintained in complete media in a T75 cm^2^ flask until they reached approximately 70% confluence. Cells were washed twice with ice-cold PBS and incubated with 750 µL radioimmunoprecipitation assay buffer (RIPA buffer) (Sigma-Aldrich, R0278, Macquarie Park, NSW, Australia) supplemented with protease and phosphatase inhibitor cocktail (Thermo Fisher Scientific, 88668, Lane Cove, NSW, Australia). Lysed cells were harvested with a scraper and centrifuged at 8000× *g* for 20 min (Hermle Centrifuge Z233 M-2, Thermo Fisher Scientific, Lane Cove, NSW, Australia) at 4 °C. Protein concentration was measured using the Pierce BCA protein assay kit (Thermo Fisher Scientific, 23225, Lane Cove, NSW, Australia) following the manufacturer’s instructions. Proteins were denatured after mixing with 2× Laemmli loading buffer (1:1) at 95 °C for 5 min in a dry bath heater (Corning LSE, Merk, Macquarie Park, NSW, Australia). Lysates (30 µg) were loaded to 10% SDS-PAGE 15-well gels (Bio-Rad. #456–1069, South Granville, NSW, Australia) and electrophoresed at 150 V for 45 min. Wet transfer by chilled Towbin buffer (25 mM Tris, 192 mM glycine, 20% (*v*/*v*) methanol (pH 8.3)) was used to transfer the protein to the PVDF membranes (Bio-Rad, #1620174, Melbourne, VIC, Australia) at 20 V for 1.5 h on mini trans-blot cell (Bio-Rad, #1703930, South Granville, NSW, Australia).

Membranes were incubated with TBS-T (0.1% Tween-20 in Tris-buffered saline) containing 5% instant skimmed milk powder (Woolworths, Wagga Wagga, NSW, Australia) for an hour. After washing twice with washing buffer TBS-T, the PVDF membranes were incubated with anti-PBR Rabbit (Abcam, ab109497, Melbourne, VIC, Australia), anti-TMEM97 (NBP1-30436, Novus Biologicals, In Vitro Technologies, Noble Park North, VIC, Australia), anti-PGRMC1 (HPA002877, Sigma-Aldrich, Macquarie Park, NSW, Australia) and anti-β-actin (ab8227, Abcam, Melbourne, VIC, Australia) and mouse anti-actin (Sigma Aldrich, A5541, Macquarie Park, NSW, Australia) at 1:1000 and 1:2000 respectively overnight at 4 °C. Next day, and the PVDF were washed twice with TBS-T and incubated with Donkey anti-Rabbit IgG H&L (HRP) (Abcam, ab16284, Melbourne, VIC, Australia) or Donkey F (ab’) 2 Anti-Mouse IgG H&L (HRP) (Abcam, ab98665, Melbourne, VIC, Australia) at 1:4000 for 1 h at room temperature. The bands’ detection was performed using Clarity Max Western ECL

### 4.5. Statistical Analysis

Statistics tools implemented in the GraphPad Prism software were used for statistical analyses. Data are represented as the mean ± standard deviation (SD) from at least three independent experiments. Student’s *t*-test was performed for pairwise comparisons. Differences between more than two groups were evaluated by one-way ANOVA followed by Dunnett’s multiple comparisons tests. For multiple cell lines and multiple treatments, 2-way ANOVA was used. For each experiment, the statistical method employed is specified in the figure legends. The asterisks in the figures denote statistical significance (* *p* < 0.05; ** *p* < 0.01; and *** *p* < 0.001).

### 4.6. shRNA Knockdown and Cell Lines Generation

Lentiviral-delivered shRNAs were constructed using Mission TRC1.5-pLKO.1-Puro series Lentiplasmids (SHCLND, Sigma-Aldrich, Macquarie Park, NSW, Australia) targeting TSPO/PBR (TRCN0000060433, CCACACTCAACTACTGCGTAT). A total of 0.4 µg Lentiviral plasmids encoding the TSPO (Sigma-Aldrich, Macquarie Park, NSW, Australia) was transformed into Competent *Escherichia coli* Top10 strain cells. Briefly, plasmids were mixed gently with Competent *E. coli* Top10 cells. The mixture was incubated for 30 min on ice. After incubation, the tube was placed in a digital dry path heater (Corning^®^ LSE™) set at 42 °C for 45 sec without shaking. Next, the tube was transferred to an ice bath for 5 min. Then, 900 µL of 2% Luria Broth (Invitrogen, 12795-027, Tullamarine, VIC, Australia) was added. The solution was mixed and incubated at 200 rpm for 60 min at 37 °C. After incubation, the solution was spread on LB agar (Sigma Aldrich, L3147, Macquarie Park, NSW, Australia) plates containing 100 µg/mL ampicillin and incubated at 37 °C overnight. Plasmid extractions were performed using the QIAprep Spin Miniprep Kit (Qiagen, 27104, Clayton VIC, Australia) according to the manufacturer’s protocol. The plasmids concentration was measured using the NanoDrop 2000 (Thermo Fisher Scientific, Lane Cove, NSW, Australia).

We generated lentivirus particles as described [61]. Briefly, HEK293 cells were transfected with the shRNA plasmids and helper plasmids using Lipofectamine 2000 (Invitrogen, 11668-027, Tullamarine, VIC, Australia). Prior to transfection, 6-well plates were treated with 50 µg/mL D-Lysine. The next day, 1 × 106 cells were seeded and incubated overnight at 37 °C. 250 µL transfection mixtures A and B were generated by incubating plasmids or Lipofectamine with antibiotic-free media for 5 min at RT. Mixture A includes 4 μg Plasmids that contain 2.5 μg TSPO shRNA, 0.75 μg Pax, 0.3 μg Rev, and 0.45 μg VSV-G [62]. Mixture B includes 8 μL of Lipofectamine incubated with 242 μL. Mixtures A and B were mixed gently and incubated for 25 min at room temperature. HEK293 cells were washed with PBS and incubated with antibiotic-free media. Then, the solution was added to the cells drop by drop with gentle shaking and incubated for 6 h at 37 °C. After incubation, the media were replaced with complete media overnight at 37 °C. Media that contain virus particles were collected every 24 h, filtered, liquated, and frozen at −80 °C.

### 4.7. siRNA Knockdown of TMEM97 in MP Cells

Sense and antisense strands for the siRNA (Sigma-Aldrich, Macquarie Park, NSW, Australia) that include:

254_GAGCUCUACCCAGUCGAGUUUAGAA

254_CUAAACUCGACUGGGUAGAGCUCUU

Control_254 GAGUCCACCGACUAGUGUUAUCGAA

Control_254 CGAUAACACUAGUCGGUGGACUCUU

For transfection, 5 × 10^5^ MIA-PaCa-2 cells were seeded overnight in 2 mL complete media containing 4 µL Lipofectamine RNAiMAX Transfection Reagent (Life Technologies, 13778030, Thermo Fisher Scientific, Lane Cove, NSW, Australia) and 10 nM of control or siRNA in a 6-well plate. The next day, cells were treated with the same combination of media and transfection reagents. On the third day, 10% of the cells were harvested to measure the expression of the TMEM97 gene by RT-PCR [20].

### 4.8. MCF7 Transfection with sh_RNA Targeting TMEM97

The procedure to develop an MCF7 cell line with reduced TMEM97 (shMCF7TMEM97 cells) was carried out according to Abate et al. [13] with minor modifications. MCF7 cells were plated at a density of 3 × 10^6^ cells in 10 mL of growth medium in 10 cm Petri dishes and incubated at 37 °C overnight. Cells were transfected with 17 µg of the pLKO.1 vector containing sh_RNA targeting TMEM97, as per standard protocol using FuGENE HD Transfection Reagent in Opti-MEM medium without serum. Vector-silencing cells were selected using puromycin. After transfection, cells were placed in a normal DMEM growth medium. After 1 day, cells were detached with trypsin/EDTA and replated into DMEM growth medium containing puromycin (2 µg/mL) and cultured for 25 days. Surviving cell clones were picked out and propagated separately in 6-cm Petri dishes in the same medium with 2 µg/mL puromycin. To suppress the reversion of the phenotype, all subsequent cell culture was carried out in DMEM growth medium as described above, supplemented with 2 µg/mL puromycin.

### 4.9. RT-PCR

RNA extraction was performed using the Total RNA mini kit (Bio-Rad, #7326820, Melbourne, VIC, Australia) following the manufacturer’s instructions. RNA concentration was measured by NanoDrop 2000 (Thermo Fisher Scientific, Lane Cove, NSW, Australia). One µg of RNA was used to synthesize cDNA using a cDNA synthesis kit (Bio-Rad, #1708890, Melbourne, VIC, Australia) in C1000 Touch™ Thermal Cycler. Primers for TMEM97 and Actin were synthesized at Monash University, Melbourne. TMEM97; Forward 5′-TACTTCGTCTCGCACATCCC-3′

Reverse 5′-TTGCTGAACTCCTGCGGGTA-3′. Actin primers were Forward 5′-ACGACAT GGAGAAAATCTG-3′ Reverse 5′-ATGATCTGGGTCATCTTCTC-3′.

RT-PCR was performed at CFX96 Touch™ Real-Time PCR Detection System using iTaq™ Universal SYBR^®^ Green Supermix (Bio-Rad, #1725121, Melbourne, VIC, Australia).

### 4.10. Flow Cytometry

Cell lines (5 × 10^5^ cells) were seeded in 6-well plates overnight. The next day, cells were incubated with either DMSO (control), 10 nM DTG (Sigma-Aldrich, Lane Cove, NSW, Australia, 207713) to block the S1R, or 4 nM RHM-1 (supplied by R.H.M.) to block the S2R. Then, the cells were incubated with 30 nM SW120 for 30 min. The cells were washed briefly with PBS three times to remove non-specific binding. The cells were trypsinized immediately and transferred to a falcon tube. Next, the cells were centrifuged at 1300× *g* for 3 min. The supernatant was discarded, and the pellets were washed twice with PBS and centrifuged. The supernatant was discarded, and the cells were resuspended in 500 µL PBS. The flow cytometry was performed by implementing a 488 nm laser for the excitation of FITC.

### 4.11. Binding Assay for SW120 and SKF 10,047

TSPO shRNA and scrambled shRNA for both MIA-PaCa-2 and HEK293T cells were incubated with 1.8 µM SKF 10,047 ligand for 1 h at 37 °C to bloke the S1R. Next, the cells were treated with 30 nM of SW120 for 30 min. The median fluorescent intensity (MFI) of the labeled cells was measured by flow beck flow cytometry. The control cells were incubated with 30 nM of SW120 for 30 min and measured by flow cytometry.

### 4.12. Proximity Ligation Assay

Coverslips were sterilized by draining in 70% EtOH, washed with PBS, and left under the UV for 30 min. Wet coverslips with PBS were placed on a 6-well plate. MP cells were seeded at <50% confluence in DMEM (Sigma-Aldrich, D5796, Macquarie Park, NSW, Australia) supplemented with 10% FBS (Sigma-Aldrich, F9423) and 1% penicillin-streptomycin (Sigma-Aldrich, P4333) at 37 °C and 5% CO_2_ in a 150i CO_2_ incubator (Thermo Fisher Scientific, Heracell, Lane Cove, NSW, Australia). After attachment to the coverslips, cells were washed twice with ice-cold PBS. Next, the cells were fixed using 3.6% paraformaldehyde in PBS pH for 10 min at room temperature. Subsequently, the cells were permeabilized with 0.1% Triton X-100 in PBS for 10 min. PLA was performed using a Duolink PLA in situ starter kit (Sigma Aldrich, DUO92101). The cells were blocked with 1 drop (~40 µL) of Duolink blocking solution for 1 h in a heated humidity chamber for 60 min at 37 °C. After blocking, the cells were incubated with diluted primary antibodies 1:100 in the Duolink antibody diluent overnight at 4 °C. The next day, the cells were washed twice with wash buffer A at room temperature for 5 min. After washing, slides were incubated with corresponding PLUS and MINUS PLA probes at 1:5 dilution in Duolink antibody diluent for 1 h at 37 °C. Slides were washed twice for 5 min with wash buffer A at room temperature and incubated with 1:40 ligase in 1x Duolink ligation buffer in high-purity water for 30 min at 37 °C. This was followed by washing twice with wash buffer A at room temperature for 5 min. For amplification, slides were incubated with 1:80 polymerase in an amplification buffer in high-purity water for 100 min at 37 °C. Slides were washed twice with wash buffer B at room temperature for 10 min, followed by a wash with 0.01x wash buffer B for 1 min at room temperature. Slides were mounted with a coverslip using a minimal volume of Duolink in Situ Mounting Medium with DAPI.

### 4.13. Confocal Microscopy Studies

MCF7 cells (3 × 10^3^/spot in 50 µL) were seeded on spot slides and allowed to recover for 48 h at 37 °C and 5% CO_2_. Afterward, the cells were incubated with 10 µM of (+)-pentazocine as a selective S1R ligand for 2 h at 37 °C and 5% CO_2_. Subsequently, the cells were treated with 5 µM of CY5-labeled S2R ligand for 1 h at 37 °C 5% CO_2_, followed by a washing step with PBS. Then, the cells were fixated with 4% paraformaldehyde at room temperature for 15 min. For staining of TSPO or PGRMC1 expression, cells were permeabilized and blocked with 0.5% Triton-X 100 and 1% bovine serum albumin (BSA), respectively, in PBS for 15 min at room temperature, followed by incubation of the primary TSPO (1:250 dilution, Abcam, cat.-nr.: ab109497, Melbourne, VIC, Australia) or PGRMC1 (1:200 dilution, MERCK, cat.-nr.:HPA064724, Sigma-Aldrich, Macquarie Park, NSW, Australia) antibodies diluted in 1% BSA 0.3% Triton-PBS overnight at 4 °C in a wet chamber. On the next day, the cells were washed three times with PBS and incubated with a 1:500 dilution of a secondary anti-rabbit AlexaFluor488-labeled antibody (Invitrogen (Waltham, MA, USA*)*, cat.-nr.: A-11034) for 1 h at room temperature in a wet chamber. After two washing steps with PBS, the cells were incubated with a solution containing DAPI (2.5 µg/mL #D9542) for 15 min at room temperature, followed by three washing steps and embedding with Vectashield (Vectashield^®^ Antifade Mounting Media, Vector Laboratories, cat.-nr.: H-1000-10, ABACUS DX, Meadowbrook, QLD, Australia). Afterward, confocal microscopy was performed on a Zeiss LSM 700 (Carl Zeiss Pty. Ltd., Macquarie Park, NSW, Australia) using a Plan-Apochromat 63x/NA 1.4/Oil lens. The pinhole size was set to 1AU. The samples were illuminated with 405 nm, 555 nm, and 639 nm lasers and 1024 × 1024 pixel images were acquired using a PMTs detector. In total, 3 pictures per spot were obtained.

## 5. Conclusions

In conclusion, our reported co-immunoprecipitation and colocalization of TMEM97/S2R with TSPO is a novel finding whose characterization requires further investigation. In addition, PGRMC1, which has already been reported as a TMEM97/S2R interactor, co-immunoprecipitates with TSPO in MP cells but not in MCF7. The different interacting partners in the two cell lines which we employed suggest that the TMEM97 interactome is context-dependent. Importantly, the PGRMC1-TSPO-TMEM complex in MP cells is sensitive to the presence of S2R/TMEM97 endogenous ligand 20S-OHC, as well as exogenous ligands, suggesting that the complex is involved in S2R cell biology. Overall, the previously unreported protein-protein interactions of these three proteins shed new light on their biology and functions and deserve further investigation that may disclose unexplored mechanisms of action and therapeutic potentials.

## Figures and Tables

**Figure 1 ijms-24-06381-f001:**
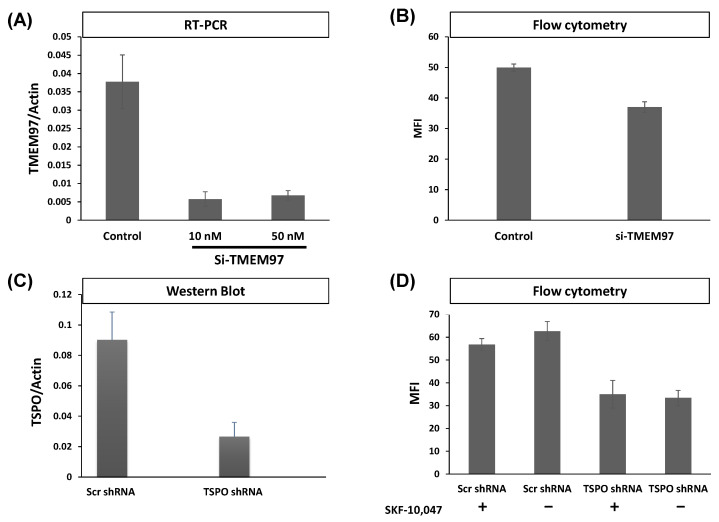
TMEM97 and TSPO attenuation both reduce S2R ligand-binding by SW120. (**A**) Gene expression by RT-PCR showing the attenuation of TMEM97 mRNA in MP cell lines. The knockdown of TMEM97 by siRNA (either 10 or 50 nM per treatment) was confirmed by more than 80% (*t*-test *p* < 0.001). (**B**) Flow cytometry assessment of SW120 binding to MP cells (*n* = 6). Median fluorescence intensity (MFI) was determined by measuring the FITC signal (FL1). Kaluza software was used to analyze the data. Attenuation of TMEM97 (10 nM siRNA) resulted in significant SW120 reduction (*t*-test *p* < 0.0001). (**C**) Detection of TSPO expression levels by Western blot after TSPO shRNA knockdown compared with scramble shRNA control. The image of the Western blot is supplied in Appendix A. Six independent clones were generated after transduction with viral particles, with all showing visible attenuation normalized to actin. A significant reduction of approximately 70% was achieved (*t*-test *p* < 0.00001). (**D**) The chart shows a flow cytometry evaluation of SW120 binding after TSPO knockdown by shRNA. MP cells were incubated with SKF-10,047 or DMSO as a negative control. All cells were incubated with 30 nM of SW120 in complete media for 30 min at 37 °C. Treatment with SKF-10.047 had no significant impact on SW120 binding (Two-way ANOVA). In contrast, attenuation of TSPO resulted in a significant reduction in the SW120 signal (Two-way ANOVA *p* < 0.0001) in both conditions. Six replicates of each cell line were tested.

**Figure 2 ijms-24-06381-f002:**
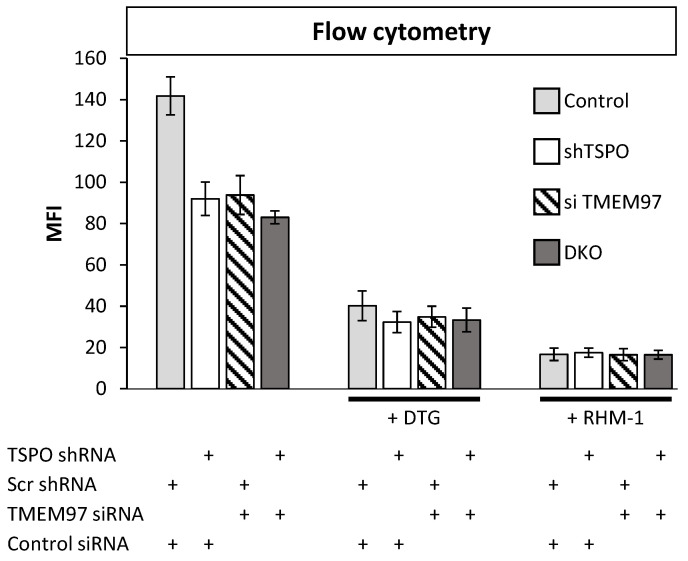
Double knockout (DKO) of TMEM97 and TSPO produced no further reduced SW120 signal in MP cells. The figure shows the first incubating condition for four differently treated cell lines. That includes Scr control cell lines, cells with an attenuated level of TSPO or TMEM97, and cells with the attenuated level of both TSPO and TMEM97 (DKO). Those cells were incubated with SW120 for 30 min at 37 °C. All treatments were significantly different from control cells (ANOVA, post hoc Tukey’s HSD, *p* < 1 × 10^−7^). The right bars show the same condition, but with the treatment of DTG or RHM-1 prior to incubation with SW120. Cells were incubated with complete media supplemented with 10 nM DTG or 4 nM of RHM-1 for 1 h at 37 °C. For DTG-treated cells (+DTG), the DTG-treated control cells were significantly different at the *p* < 0.0001 level from the untreated control cells. All DTG-treatment conditions (shTSPO, si TMEM97, and DKO) were significantly different from DTG-treated control at the *p* < 0.005 level (*t*-test on + DTG cells only). All RHM1-treated cells (+ RHM-1) were significantly different from respective untreated cells at the *p* < 0.001 level (*t*-test on + RHM-1 cells only). There was no significant difference between Control, shTSPO, siTMEM97, or DKO for RHM-1 and DTG-treated cells (*t*-test on + DTG or + RHM-1 treated cells).

**Figure 3 ijms-24-06381-f003:**
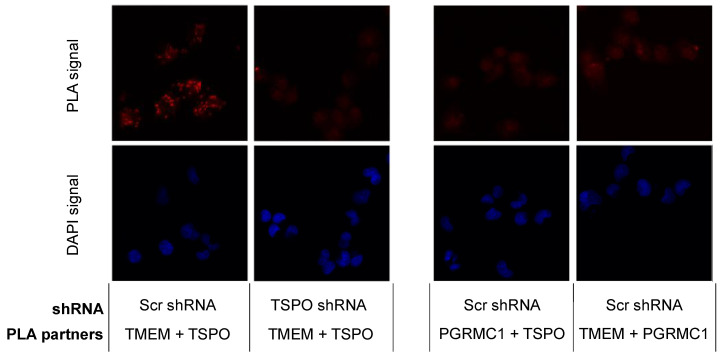
TSPO is colocalized with SW120 receptor S2R in MP cells. The left panels represent PLA results for TMEM97 (TMEM) and TSPO pairing. The positive signal was observed compared with the negative signal after attenuating TSPO. The right panels show two negative PLA signals for TMEM97 + PGRMC1 and PGRMC1 + TSPO. Those negative observations were similar to a negative control without primary antibodies. Red corresponds to the signal amplified by the Duolink PLA reaction described in Section 4.12. Blue represents DAPI staining of DNA. Each discrete blue patch represents one cell nucleus (imaged at 20× magnification).

**Figure 4 ijms-24-06381-f004:**
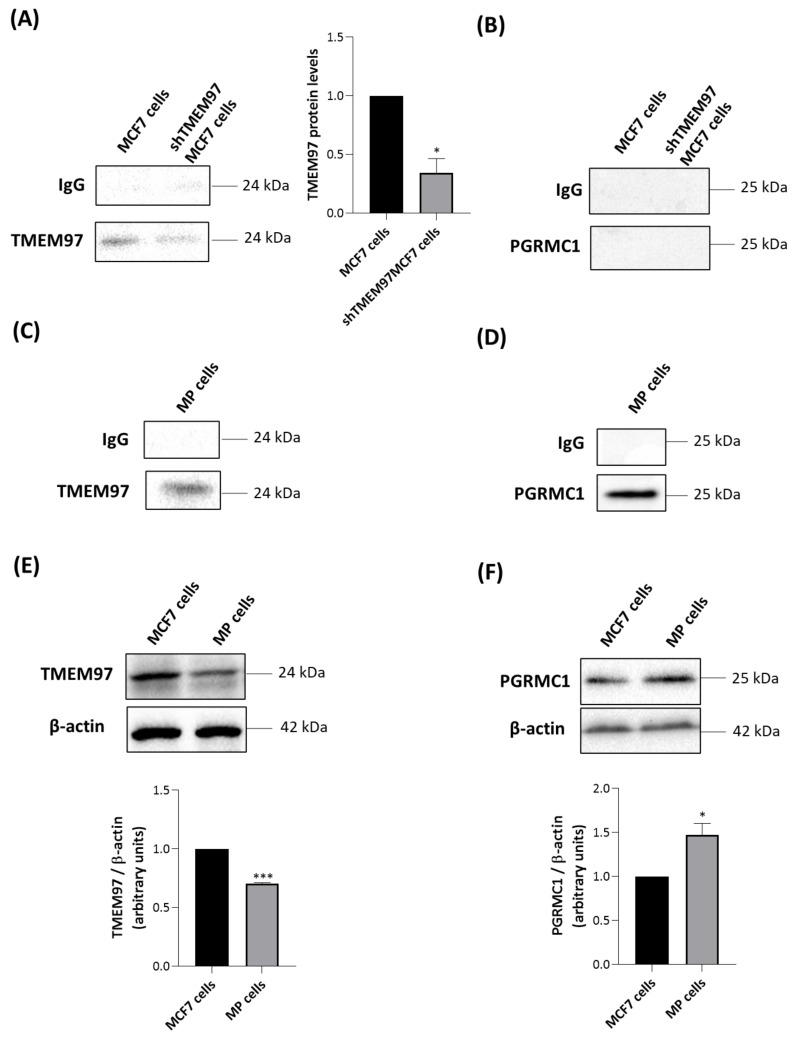
TSPO co-immunoprecipitates only with TMEM97 in MCF7 cells and with both TMEM97 and PGRMC1 in MP cells. MCF7 and shTMEM97MCF7 cells were immunoprecipitated with an antibody directed against TSPO and analyzed by Western blot with anti-TMEM97 (**A**) or anti-PGRMC1 (**B**) antibodies. TMEM97 protein levels in shTMEM97MCF7 cells were normalized versus proteins in MCF7 cells. MP cells were immunoprecipitated with an antibody directed to TSPO and analyzed by Western blot with anti-TMEM97 (**C**) or anti-PGRMC1 (**D**) antibodies. IgG is the negative control. Western blot analyses were performed to evaluate the protein levels of TMEM97 (**E**) and PGRMC1 (**F**) in MCF7 MP cells. In (**E**,**F**) protein levels were normalized against β-actin. Western blotting data are representative of at least three independent experiments. Graph bars were used to represent the means ±SD (error bars). Statistical significance of the differences was evaluated by using Student’s *t*-test (* *p* < 0.05, *** *p* < 0.001).

**Figure 5 ijms-24-06381-f005:**
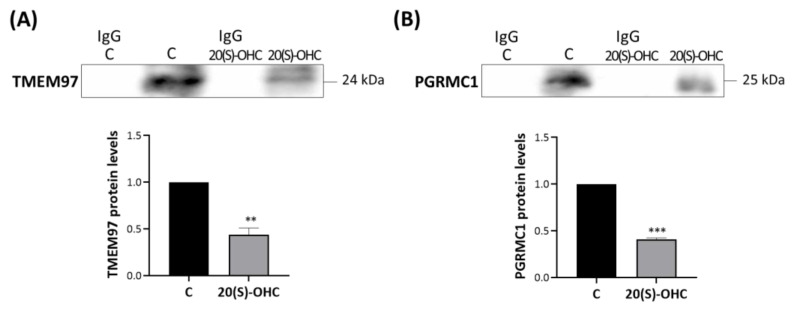
20S-OHC reduces levels of both TMEM97 and PGRMC1 co-immunoprecipitated with TSPO. MP cells were treated for 24 h with 20S-OHC, immunoprecipitated with an antibody directed against TSPO, and the immune pellets were analyzed by Western blot with anti-TMEM97 (**A**) or anti-PGRMC1 (**B**) specific antibodies. IgG is the negative control for the anti-TSPO experimental treatment immunoprecipitation. Western blotting data are representative of at least three independent experiments. TMEM97 (**A**) and PGRMC1 (**B**) protein levels in 20S-OHC treated cells were normalized versus proteins in untreated cells (C) and reported in the respective graph bars as means ± SD (error bars). Statistical significance of the differences was evaluated by using Student’s *t*-test (** *p* < 0.01, *** *p* < 0.001).

**Figure 6 ijms-24-06381-f006:**
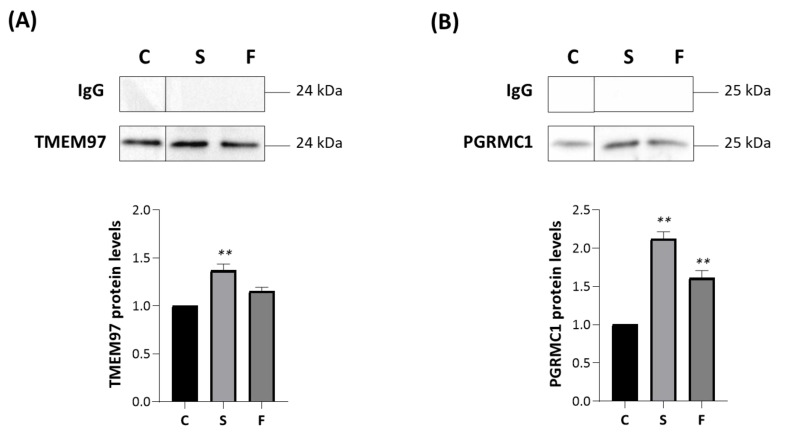
Siramesine oxalate salt and FA10 hydrochloric salt affect the associations TSPO-TMEM97 and TSPO-PGRMC1. MP cells were treated for 2 h with siramesine oxalate salt (S) and FA10 hydrochloric salt (F), immunoprecipitated with an antibody directed against TSPO and analyzed by Western blot with anti-TMEM97 (**A**) or anti-PGRMC1 (**B**) specific antibodies. IgG is the negative control for the TSPO antibody. Western blotting graphs depict the results of at least three independent experiments. TMEM97 and PGRMC1 protein levels in treated cells were normalized versus proteins in untreated cells (C) and reported in the respective graph bars (lower panels) as means ± SD (error bars). Statistical significance of the differences was evaluated by using one-way ANOVA followed by Dunnett’s multiple comparisons test (** *p* < 0.01).

**Table 1 ijms-24-06381-t001:** TSPO and Sigma-2 receptors expression in different cell lines.

Cell Lines	B_max_, pmol/mg of Protein	
	TSPO	TMEM97/S2R
C6 rat glioma	3.60	2.15
MCF7	1.02	1.795
shMCF7TMEM97	2.06	0.891
LoVo	3.43	0.54
Mia PaCa-2 (MP)	7.30	1.14

## Data Availability

All data have been reported herein or in the references cited.

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
