# Peer review of "Sigma-2 Receptor Ligand Binding Modulates Association between TSPO and TMEM97"

_ijms, 2023, doi:10.3390/ijms24076381_

Round 1
Reviewer 1 Report
“Evidence for Functional Association between TSPO and TMEM97 Regulated by Sigma-2 Receptor Ligand Binding in MCF7 and MIA PaCa-2 Cells” by Thejer et al. is a very well written manuscript. The introduction was relevant and summarizes the rationale behind this paper. While the results were simple and straightforward, they are also clearly presented with good controls in place for S1R/SR2 specificity and antibody/model validations. I recommend publication after some minor revisions. Please refer to some points and editorial suggestions below.
Specific points:
-The rationale behind choosing MCF7 and MP cell lines for further experiments should be mentioned earlier rather than at the discussion…perhaps at the saturation analyses section (section 2.4) or introduction. It wasn’t clear in section 2.4 why MCF7 and MP cells were chosen.
-The order for section 2.3 (PLA) and section 2.4 is a bit strange. If different cell lines were screened for TSPO and TMEM97, shouldn’t it go earlier? Also, PLA is arguably a stronger evidence of TMEM/TSPO interaction given the proximity prerequisite, so the level of evidence for PLA is actually higher than Co-IP (which could connected via a large complex). I would suggest a new order: section 2.4 (Saturation analyses, cell line selection); 2.5, 2.6 (Co-IP), then 2.3, followed by 2.7 and 2.8. This way, Section 2.3, 2.7, 2.8 are all done in MP cells as well. It is ok to focus on MP cells later but to give a rationale (e.g. TSPO is higher therefore it is easier to detected Co-IP signals).
-I recommend slightly tweaked and potentially shortened the title, as it is too wordy. Majority of the results were from MP cells while there are only some CoIP data from MCF7. The specific cell lines may not be necessary to be in the title. Also, the “functional association” seems like a bit of a stretch. The data presented here definitely implicate that there could be some functional role but the data here are mostly physical evidence. As mentioned in discussion, more work needs to be done to confirm function of such interaction. I would recommend checking the rest of the manuscript to be careful with stating "functional" without explaining further. “Evidence for” also weakens the title. I would suggest something simple like the following: “Sigma-2 receptor ligand binding modulates association between TSPO and TMEM97.”
Minor or editorial:
Line 94: State the actual distance range of PLA detection. I believe it is <40nm or so.
Figure 1A: Is the 10nM and 50nM the siRNA concentration used? Please clarify that.
Figure 1A: Asterisk brackets can be used to show statistical significant differences.
Section 2.2/ Figure 2B: What is the limit of detection for the MFI? Is it possible that MFI limit of detection is reached for that selected dose of RHM-1? Would you expect any changes in pattern if a lower dose of RHM-1 is used (with doses at nanomolar, it looks like it has a fairly high potency)?
Figure 2A-B: Suggest combining A and B bar graphs into one. Non-treated bar next to RHM-1-treated bar.
Section 2.3/Figure 3: Images are too blurry. The resolution needs to be fixed. Also, is there some way to quantify these images (e.g. average pixel intensity, foci per cell, etc.)?
Section 2.4: Please briefly elaborate a bit on the saturation analyses here. It was not immediate clear that it is a dose response curve by radioligand until I digged through the methods and supplementary.
Table 1: The shTMEM97 subline of MCF7 has a higher level TSPO? Isn’t that interesting? Could be discussed more. Maybe a compensatory increase?
Line 347: grammar change - suggest removing “Indeed, the effect of” and start with “The administration of….” And remove the first comma on line 348
Line 350: Suggest changing “We know not whether” to “It is unknown whether”
Author Response
Please, see the attachment.

Reviewer 2 Report
This manuscript studied the S2R ligand binding proteins in MCF7 and MiaPaca cell lines. They found that treatment with the TMEM97 ligand 20-(S)-hydroxycholesterol reduced co-immunoprecipitation of both TMEM97 and PGRMC1 in immune pellets of immunoprecipitated TSPO in MP cells. This manuscript increases our knowledge to some extend in the field, however, there are still gaps need to be filled to make the manuscript stronger.
Specific comments:
1. Fig1A, description of 10nM and 50nM of what will help to better understand.
2. Is there western blot data available to show the knockdown efficiency of TMEM97 and TSPO?
3. Introduction of SW120 binding will help the audience to better understand.
4. Is S2R level higher in MCF7 and MiaPaca2 comparing to other breast cancer cells and pancreatic cancer cells?
5. Does siTMEM97 affect the S2R translocation?
6. Does siTMEM97 affect the heme and sterol metabolism?
7. Is there a second cell line beyond MCF7 and MiaPaca had the same trend? Is there a second siRNA available to use and show the same trend?
Author Response
Please, see the attachment.

Round 2
Reviewer 2 Report
The authors answered my questions. No further comments.
Author Response
Dear Reviewer,
We thank you for considering the changes we made and the appreciation of our manuscript.
Sincerely,
Carmen Abate